# GIS-Based Gully Erosion Susceptibility Mapping: A Comparison of Computational Ensemble Data Mining Models

**Viet-Ha Nhu [1,2], Saeid Janizadeh [3], Mohammadtaghi Avand [3], Wei Chen [4], Mohsen Farzin [5], Ebrahim Omidvar [6], Ataollah Shirzadi [7], Himan Shahabi [8,9], John J. Clague [10], Abolfazl Jaafari [11], Fatemeh Mansoorypoor [12], Binh Thai Pham [13,*], Baharin Bin Ahmad [14] and Saro Lee [15,16,*]**

[1] Geographic Information Science Research Group, Ton Duc Thang University, Ho Chi Minh City 758307, Vietnam; nhuvietha@tdtu.edu.vn
[2] Faculty of Environment and Labour Safety, Ton Duc Thang University, Ho Chi Minh City 758307, Vietnam
[3] Department of Watershed Management Engineering, College of Natural Resources, Tarbiat Modares University, Tehran, P.O. Box 14115-111, Iran; Janizadeh.saeed@gmail.com (S.J.); Mt.avand70@gmail.com (M.A.)
[4] College of Geology & Environment, Xi'an University of Science and Technology, Xi'an 710054, China; chenwei.0930@163.com
[5] Department of Forestry, Range and Watershed Management, Faculty of Agriculture and Natural Resources, Yasouj University, Yasouj 75918-74934, Iran; m.farzin@yu.ac.ir
[6] Department of Rangeland and Watershed Management, Faculty of Natural Resources and Earth Sciences, University of Kashan, Kashan 87317-53153, Iran; ebrahimomidvar@kashanu.ac.ir
[7] Department of Rangeland and Watershed Management, Faculty of Natural Resources, University of Kurdistan, Sanandaj 66177-15175, Iran; a.shirzadi@uok.ac.ir
[8] Department of Geomorphology, Faculty of Natural Resources, University of Kurdistan, Sanandaj 66177-15175, Iran; h.shahabi@uok.ac.ir
[9] Board Member of Department of Zrebar Lake Environmental Research, Kurdistan Studies Institute, University of Kurdistan, Sanandaj 66177-15175, Iran
[10] Department of Earth Sciences, Simon Fraser University, Burnaby, BC V5A 1S6, Canada; john_clague@sfu.ca
[11] Research Institute of Forests and Rangelands, Agricultural Research, Education, and Extension Organization (AREEO), Tehran P.O. Box 64414-356, Iran; jaafari@rifr-ac.ir
[12] Data Mining Laboratory, Department of Engineering, College of Farabi, University of Tehran, Tehran 37181-17469, Iran; Fatemehmansoorypoor@gmail.com
[13] Institute of Research and Development, Duy Tan University, Da Nang 550000, Vietnam
[14] Department of Geoinformation, Faculty of Built Environment and Surveying, Universiti Teknologi Malaysia (UTM), Johor Bahru 81310, Malaysia; baharinahmad@utm.my
[15] Geoscience Platform Research Division, Korea Institute of Geoscience and Mineral Resources (KIGAM), 124 Gwahak-ro, Yuseong-gu, Daejeon 34132, Korea
[16] Department of Geophysical Exploration, Korea University of Science and Technology, 217 Gajeong-ro, Yuseong-gu, Daejeon 34113, Korea
*   Correspondence: phamthaibinh2@duytan.edu.vn (B.T.P.); leesaro@kigam.re.kr (S.L.); Tel.: +82-42-8683-057 (S.L.)

**Abstract:** Gully erosion destroys agricultural and domestic grazing land in many countries, especially those with arid and semi-arid climates and easily eroded rocks and soils. It also generates large amounts of sediment that can adversely impact downstream river channels. The main objective of this research is to accurately detect and predict areas prone to gully erosion. In this paper, we couple hybrid models of a commonly used base classifier (reduced pruning error tree, REPTree) with AdaBoost (AB), bagging (Bag), and random subspace (RS) algorithms to create gully erosion susceptibility maps for a sub-basin of the Shoor River watershed in northwestern Iran. We compare

the performance of these models in terms of their ability to predict gully erosion and discuss their potential use in other arid and semi-arid areas. Our database comprises 242 gully erosion locations, which we randomly divided into training and testing sets with a ratio of 70/30. Based on expert knowledge and analysis of aerial photographs and satellite images, we selected 12 conditioning factors for gully erosion. We used multi-collinearity statistical techniques in the modeling process, and checked model performance using statistical indexes including precision, recall, F-measure, Matthew correlation coefficient (MCC), receiver operatic characteristic curve (ROC), precision–recall graph (PRC), Kappa, root mean square error (RMSE), relative absolute error (PRSE), mean absolute error (MAE), and relative absolute error (RAE). Results show that rainfall, elevation, and river density are the most important factors for gully erosion susceptibility mapping in the study area. All three hybrid models that we tested significantly enhanced and improved the predictive power of REPTree (AUC=0.800), but the RS-REPTree (AUC= 0.860) ensemble model outperformed the Bag-REPTree (AUC= 0.841) and the AB-REPTree (AUC= 0.805) models. We suggest that decision makers, planners, and environmental engineers employ the RS-REPTree hybrid model to better manage gully erosion-prone areas in Iran.

**Keywords:** gully erosion; watershed management; machine learning; hybrid models; GIS; Iran

---

## 1. Introduction

A global problem that seriously threatens soil and water resources is soil erosion [1–3]. Gully erosion affects soil productivity, can trigger debris landslides and debris flows [4,5], and—if sufficiently severe—can cause an undesirable buildup of sediment in waterways, reservoirs, and ponds [6,7]. Gullies are deep erosional channels on slopes and are commonly a product of ephemeral runoff during periods of heavy rainfall. They provide pathways for water and sediment transport from the upper to lower parts of watersheds. In some catchments, as much as one-third to one-half of the total sediment output is a product of gully erosion [8,9], and gully erosion constitutes 10 to 94 percent of erosion at the watershed scale [9,10]. Gully networks also lower the water table in eroded areas, reducing soil moisture and potentially lowering crop yields on the damaged terrain.

Identifying areas that are susceptible to gully erosion can help land-use managers and planners maintain soil and water resources [10]. Dealing with gullies after they begin to form is difficult and expensive, thus it is better to plan and implement preventative and protective schemes before erosion begins [11].

Past attempts to identify slopes susceptible to gully erosion have focused on topographic thresholds. However, models that use only topographic thresholds typically fail to identify locations sensitive to gully erosion [12,13]. They de-emphasize or ignore land-use, hydrological, climatic, and other environmental factors that have key roles in gully erosion, and do not consider the rapid growth of gully systems once they have initiated [14–17].

Scientists have used a variety of computational data mining methods and models in natural hazard research, including studies of floods [18–28], wildfire [29], sinkholes [30], droughtiness [31,32], earthquakes [33,34], land/ground subsidence [35,36], groundwater [21,37–44], and landslides [22,45–72]. These methods extract related patterns in historical data to predict future events [73]. Data mining methods used to predict gully erosion include logistic regression (LR) [2,30,74–77], artificial neural network (ANN) [20,48,78–80], random subspace (RS) [48,62,81], maximum entropy (ME) [82], artificial neural fuzzy system (ANFIS) [56,83–86], support vector machine (SVM) [18,59,73], fuzzy analytical network (FAN) [37], multi-criteria decision analysis (MCDA) [87,88], evidential belief function (EBF) [88,89], classification and regression tree (CART) [90,91], random forest (RF) [39,52,92–94], rotation forest (RoF) [95], weights of evidence (WofE) [96], frequency ratio (FR) [28,97], BFTree for gully headcut [81], boosted regression [24], ADTree, RF-ADTree [73,76,98], and naive Bayes tree (NBTree) [67].

Accurate gully erosion susceptibility maps are required to predict, control, and mitigate gully formation. This need has led researchers to apply and test a wide variety of data mining methods in gully-prone areas. This study uses three hybrid models—Ada-REPTree, Bag-REPTree, and RS-REPTree—to prepare gully erosion hazard zoning maps for the Rabat Turk watershed in northwestern Iran and to compare the results with those obtained using other models. The study area has an arid to semi-arid climate, a limited vegetation cover, and easily eroded bedrock, all of which make it susceptible to gully erosion.

## 2. Materials and Methods

### 2.1. Study Area

The Rabat Turk watershed is located between Markazi and Isfahan provinces in northwestern Iran (Figure 1). It is one of the catchments of the Shoor River watershed and has an area of about 242 km². The lowest elevation in the watershed is 1807 m above sea level (a.s.l); its maximum elevation is 2723 m a.s.l. The climate is arid and semi-arid, with average annual rainfall of 213 mm. Precipitation is seasonal, with about 80% of the annual rainfall falling between December and early April [93]. Most of the catchment is bare land, although some areas support agriculture and domestic animals. Gullies are concentrated in the northern part of the watershed, and most are active [93] (Figure 2).

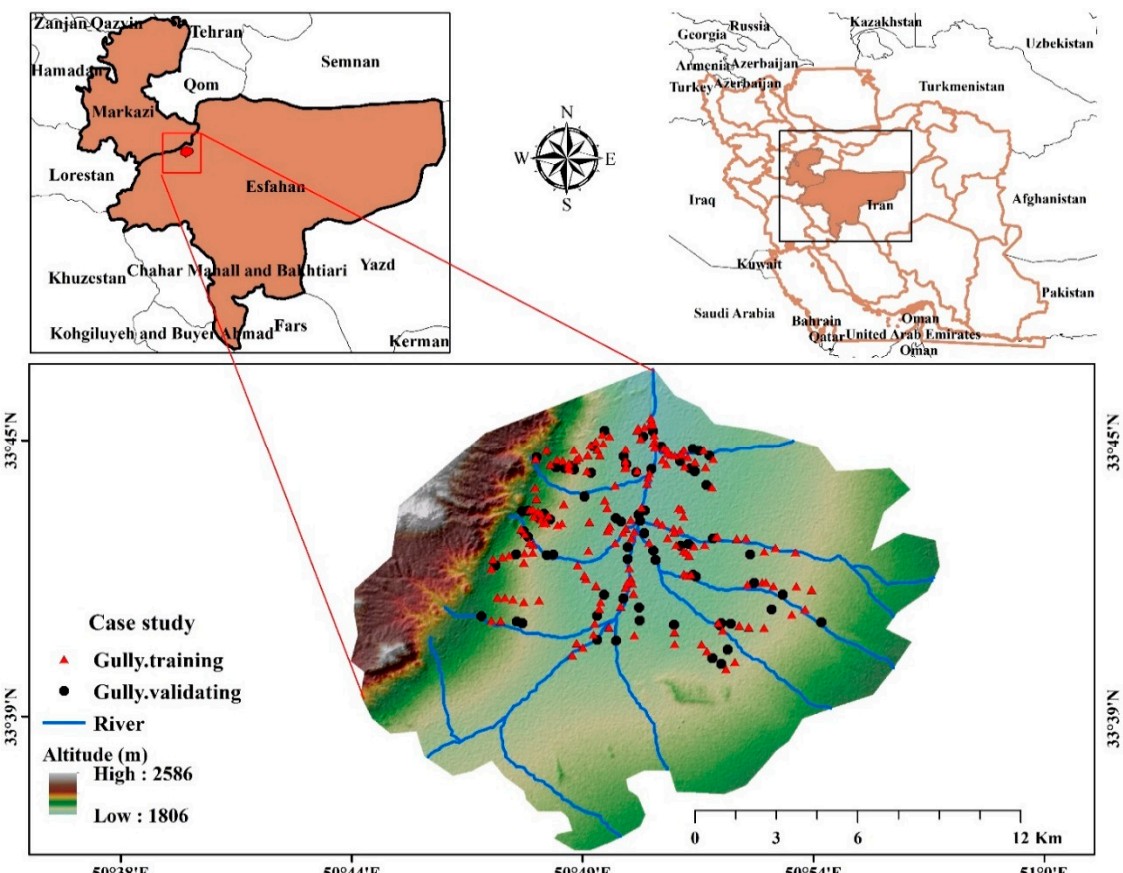

**Figure 1.** Location of the study area and model training and validating gullies.

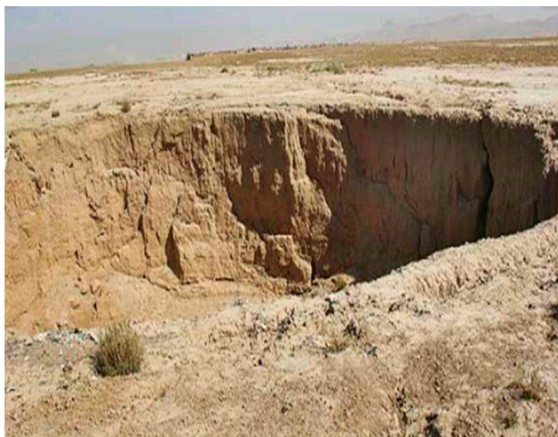
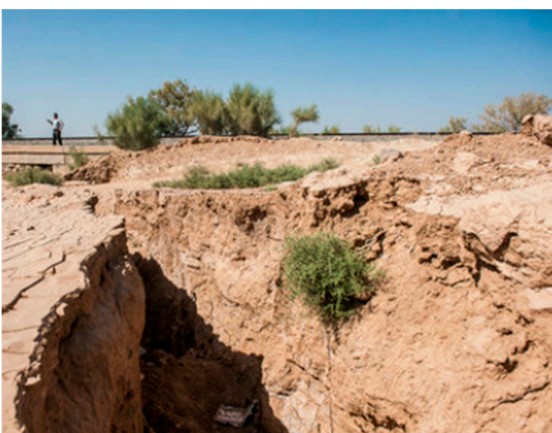

**Figure 2.** Examples of gully erosion in the study area.

## 2.2. Methodology

A flowchart for the methodology used in this study is shown in Figure 3. The methodology involves the following steps: (1) preparing a gully erosion inventory map; (2) determining the appropriate gully erosion conditioning factors (factor ranking and selection); (3) modeling gully erosion susceptibility using REPTree and its ensembles—AdaBoost, bagging, and random subspace algorithms; (4) assessing the goodness-of-fit and prediction accuracy of the models, (5) generating flood susceptibility maps using a base classifier and its ensembles, and (6) assessing the goodness-of-fit and prediction accuracy of the maps.

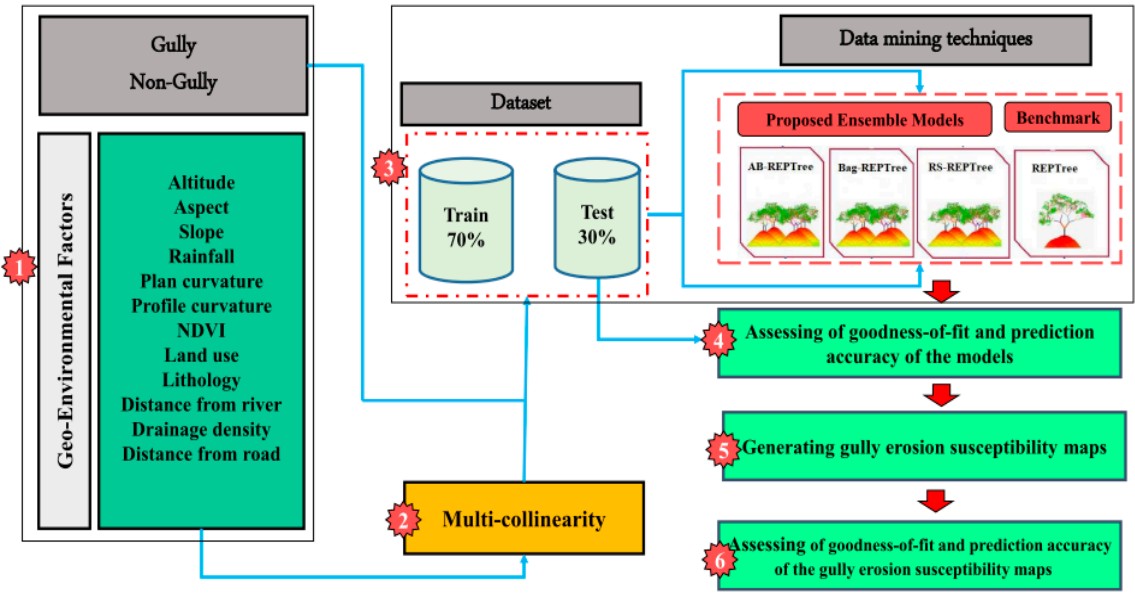

**Figure 3.** Flowchart of the study.

### 2.2.1. Gully Inventory Map

Accurately predicting and modeling gully erosion susceptibility requires a high-quality gully erosion map is essential, which thus must be carefully prepared. We obtained an inventory map with 242 gully locations from the Administration of Natural Resources of Markazi Province. The gullies were mapped from aerial photographs and satellite images and were confirmed in the field. Typically, gullies in the study area have concave and vertical heads, indicating that they are active. Longitudinal

profiles are typically straight to convex, but gully widths differ greatly. Gullies on agricultural land commonly have V-shaped cross-sections, whereas those on rangeland more commonly are U-shaped.

Depending on map scale, a gully may be considered a point or a polygon. Most authors who have studied gully erosion consider the heads of gullies to be gully locations [76,99,100], because gully heads are the sources of much of the sediment carried by the gully channels and delivered to the fluvial system below [101,102]. However, some researchers have used grid cells to create gully polygons to prepare gully erosion susceptibility maps [92,103,104], whereas others have converted gully polygons to points using 'feature to point' tool in ArcGIS software [105]. However, an active gully is a dynamic landform, and its head moves landward over time as erosion proceeds. A gully consists of three parts: its head, the main channel, and its end point. For long gullies, we used these three points to define their locations. For short gullies, we considered only the head location point. For this study, we randomly selected 242 non-gully locations in the study area. We randomly chose 70% (169) of the mapped gullies to construct the model for gully erosion; the remaining 30% (73) were used to evaluate the predictive performance of model (Figure 3).

### 2.2.2. Gully Conditioning Factors

Gully erosion is a complex process that results from the interplay of numerous factors [106,107]. After reviewing gully erosion literature and considering local conditions and available data, we selected 12 topographic, hydrological, geological, and anthropogenic factors for inclusion in the modeling process.

The topographic factors chosen for this study are elevation, aspect, slope gradient, plan curvature, and profile curvature. The hydrological parameters are distance to rivers and drainage density. We extracted topographic and hydrological factors from a digital elevation model (DEM) obtained from ALOS PALSAR (Phased Array Type L-band Synthetic Aperture Radar) data, with a cell size 12.5 × 12.5 m (http://www.eorc.jaxa.jp/ALOS/en/aw3d30) and prepared in ArcGIS 10.3 [93].

The elevation map has four classes (1800–2000, 2000–2200, 2200–2400, and >2400 m a.s.l) (Figure 4a). The highest gully frequency ratio (FR) is associated with the 1800-2000 m class (FR ratio = 1.16). The gully aspect map (Figure 4b) has nine classes, and the highest FR values are in the east, northeast, and southeast aspect classes, with values of, respectively, 1.30, 1.17, and 1.13. The slope gradient map has five classes: 0–5%, 5–10%, 10–20%, 20–30%, and >30% (Figure 4c). The 5–10% class has the highest FR value (1.23). Plan curvature was categorized as convex, flat, and concave forms (Figure 4d). Most gully erosion in the study area occurs in areas mapped as flat (FR = 1.09). There are three classes of profile curvature (< −0.35, −0.35–0.25, and >0.25) (Figure 4e). The <−0.35 class has the highest FR value (1.18).

Hydrological factors (distance from river, drainage density, and rainfall) were extracted from the stream network in the DEM using the Arc Hydro, Euclidean Distance, and Line Density in Spatial Analysis tools in ArcGIS 10.3 [108]. Distance-from-river classes are 0–500, 500–1000, 1000–2000, 2000–3000, and >3000 m (Figure 4f). Gully erosion in the study area is greatest near rivers, and thus the 0-500 m class has the highest FR (1.63). The drainage density map has five classes: 0–0.24, 0.24–0.64, 0.64–1.06, 1.06–1.62, and 1.62–2.46 km/km$^2$ (Figure 4g). Gully erosion and drainage density are positively correlated; therefore the 1.62–2.46 class has the highest FR value (4.32) and the 0–0.24 class has the lowest FR value (0.52). Annual rainfall data for the study area were obtained for the period 1984–2014 from Robat Turk watershed weather stations operated by the Iran Meteorological Organization. Based on previous related research [76], gully erosion and rainfall are inversely correlated. The rainfall data were interpolated using the inverse distance weighting (IDW) interpolation tool in ArcGIS 10.3 and placed into three classes: 148–159, 159–171, and 171–192 mm (Figure 4l). The largest and smallest number of gullies in the study area are in, respectively, the 148–159 mm (FR = 2.15) and 171–192 mm (FR = 0 classes).

Bedrock lithology is an important factor in gullying [8], and eight types were extracted from a 1:100,000-scale geological map using ArcGIS 10.3 (Figure 4h). The highest and lowest FR values belong to, respectively, the gypsum (Ekgy) class (4.43) and regional metamorphic rocks (pCmt2) (0.08).

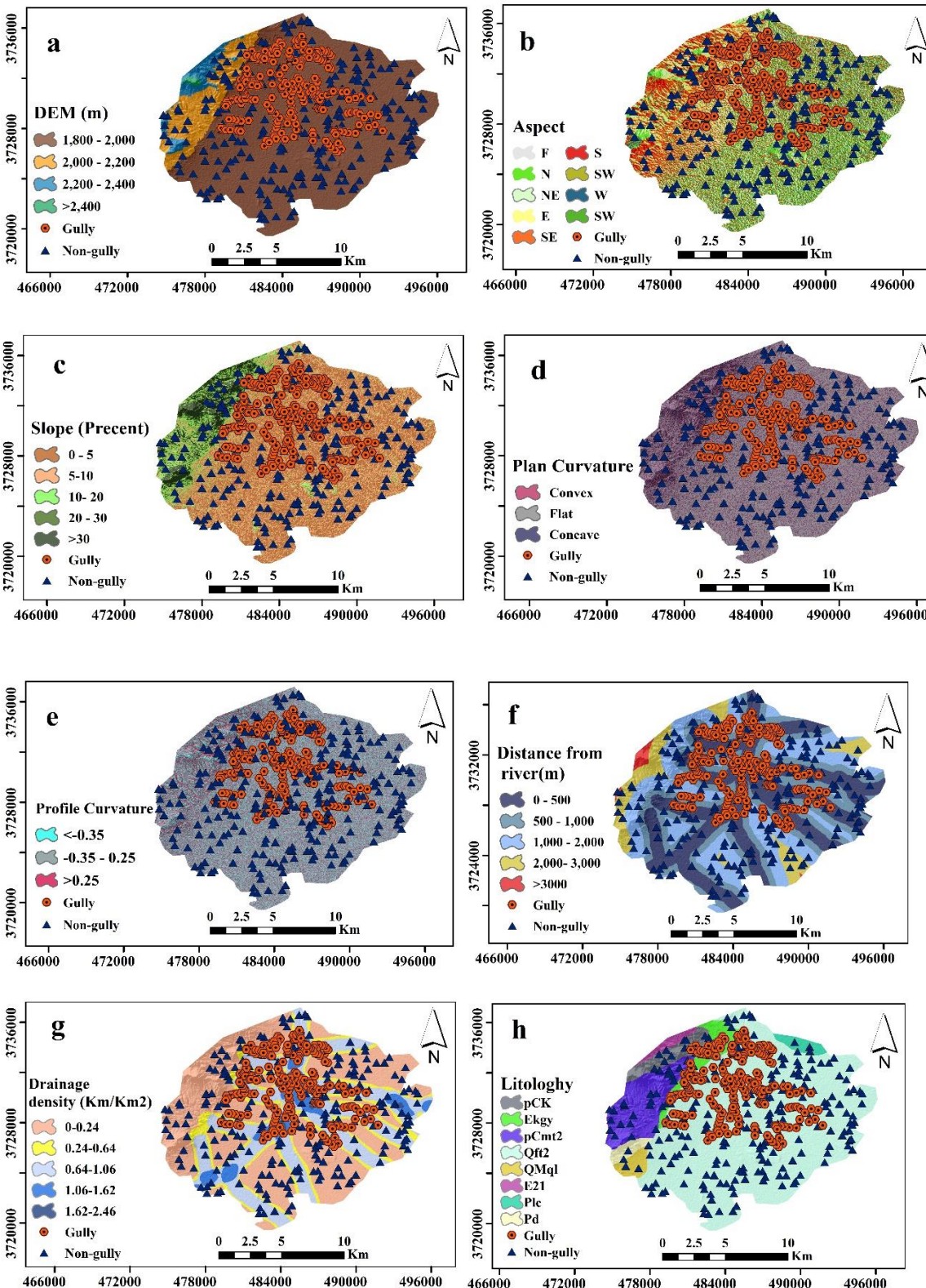

**Figure 4.** *Cont.*

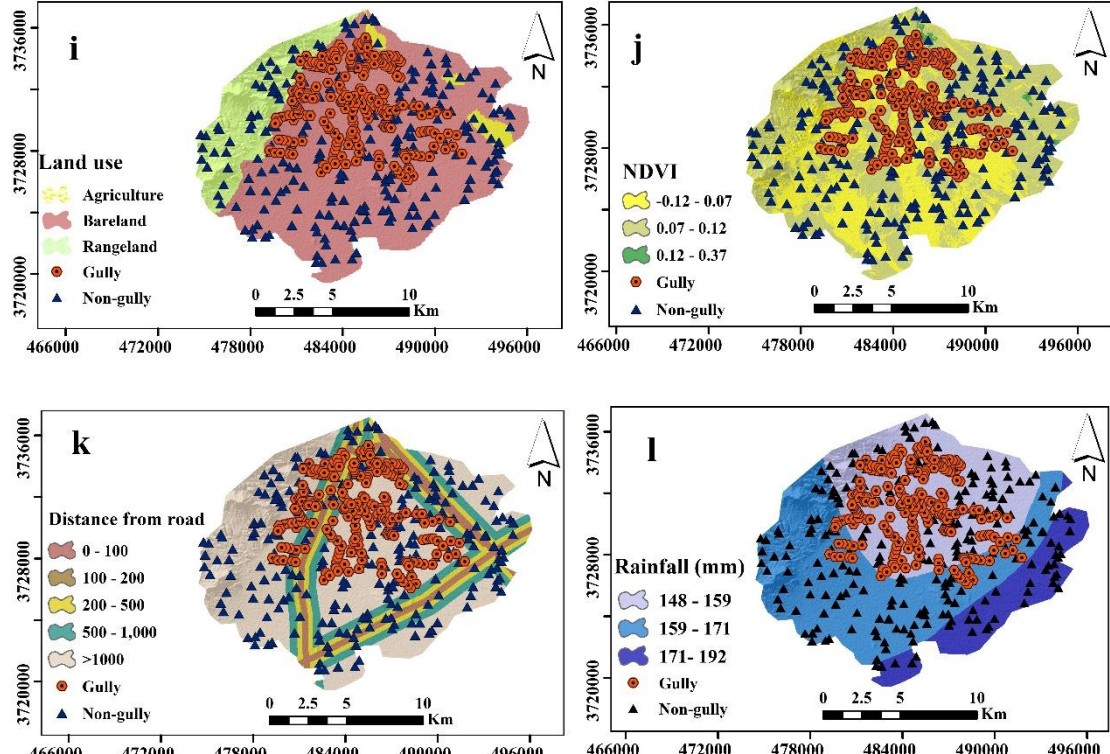

**Figure 4.** Spatial database for gully susceptibility analysis. (**a**) Elevation, (**b**) aspect, (**c**) slope, (**d**) plan curvature, (**e**) profile curvature, (**f**) distance from river, (**g**) drainage density, (**h**) lithology, (**i**) land use, (**j**) NDVI, (**k**) distance from road, (**l**) rainfall.

Changing land use, for example deforestation and grazing, is an important cause of soil erosion [76]. For the current study, land use was inferred from Landsat 8 (OLI) satellite imagery and analyzed and processed with the ENVI 5.4 software. The land-use map includes three classes—agriculture, bare land, and rangeland (Figure 4i). Most gullies in the study area are found in the bare land class (FR = 1.21), and lowest number are in the rangeland class (FR = 0.62).

The incidence of gully erosion is greatest in areas with limited vegetation cover. A normalized difference vegetation index (NDVI) map of the study area was generated in ArcGIS 10.3 from Landsat 8 imagery acquired on 15 June 2017. This map is based on the formula (NIR-Red)/(NIR+Red), where NIR (near-infrared) is band 5 and Red is band 4 of the Landsat 8 imagery. The map includes three NDVI classes: −0.12–0.07, 0.07–0.12, and 0.12–0.37 (Figure 4j). The 0.12–0.37 map class has the largest number of gullies.

Roads also affect gully erosion, as they intercept and concentrate overland flow [109,110]. This factor is represented by distances of gully and non-gully sites from roads, which were determined by vectorizing topographic maps and then transforming the data to a raster map using the Euclidean Distance tool in Arc GIS 10.3. Five classes were defined: 0–100, 100–200, 200–500, 500–1000, and >1000 m (Figure 4i). The largest and smallest number of gullies in the study area are, respectively, in the 200–500 m (FR = 1.10) and 0–100 m (FR = 0.73) classes.

### 2.2.3. Gully Erosion Susceptibility Modeling

In this study, we prepared gully susceptibility maps using REPTree as a base classifier and AdaBoost, bagging, and random subspace in ensemble models. The following subsections briefly describe the four ensemble models.

AdaBoost (AB)

AdaBoost (adaptive boosting) was the first boosting algorithm used for binary classification [111] and is a starting point for understanding the concept of boosting. AdaBoost free users from the complexities involved in detecting and choosing parameters.

The steps of the AdaBoost algorithm can be summarized as follows:

First, each data point is calculated as

$$w(x_i, y_i) = \frac{1}{n}, \ i = 1, \dots, n \tag{1}$$

The obtained weights are updated after each step.

Second, a basic classifier $C_b(X_i)$ is built from a training set and is applied to each training sample. The error of this classifier $\varepsilon_b$ is calculated as

$$\varepsilon_b = \sum_{i=1}^{n} w_b(i) \xi_b(i) \ where \ \xi_b(i) = \begin{cases} 0 & C_b(x_i) = y_i \\ 1 & C_b(x_i) \neq y_i \end{cases} \tag{2}$$

The new weight for each iteration is

$$w_{b+1}(i) = w_b(i) . exp(\alpha_b \xi_b(i)) \tag{3}$$

where $\alpha_b$ is a constant that is calculated from the error of the classifier in each iteration

$$\alpha_b = ln((1 - \varepsilon_b)/\varepsilon_b) \tag{4}$$

The calculated weights in each iteration are generally normalized, and their sum is one.

This process is repeated in every step for b = 1, 2, 3, ... , B, and then the ensemble classifier is built as a linear combination of the single classifiers weighted by the corresponding constant $\alpha_b$:

$$C(x) = sign\left(\sum_{b-1}^{B} \alpha_b C_b(x)\right) \tag{5}$$

Bagging (Bag)

Bagging is an ensemble learning method introduced by Breiman [112]. It creates parallel diverse classifiers that are then coupled. Specifically, each bootstrap sample dataset is generated by randomly drawing, with replacement, N instances (N is the size of the original training datasets). Then, a classifier $C_i$ is built from each bootstrap sample $B_i$, and $C^*$ is built from $C_1, C_2, \dots, C_T$. Bagging output is the class that is most often predicted by its sub-classifiers.

This algorithm can be summarized as follows:

Input: training set *S*, inducer *T*, integer *T* (number of bootstrap samples)

(1)  for $i = 1$ to *T* {
(2)  $S^i$ = bootstrap sample from S (sample with replacement)
(3)  $C_i = T(S^i)$
(4)  }
(5)  $C^*(x) = \underset{y \in Y}{\mathrm{argmax}} \sum_{i=C_i(x)=y} 1$
(6)  Output: classifier $C^*$

Random Subspace

The random subspace (RS) method [113] is an ensemble classifier technique in which each training sample is defined as a p-dimensional vector $\mathbf{X}_i = (x_{i1}, x_{i2}, \ldots, x_{ip})$ and r<p features are randomly selected from the p-dimensional dataset X in each iteration. Classifiers then are built into the random subspaces and aggregated through majority voting.

The RS algorithm can be summarized as follows:

(1) Repeat for $b = 1, 2, \ldots, B$:

    (a)    Select a r-dimensional random subspace $\widetilde{X}^b$ from the original p-dimensional feature space.

    (b)    Construct a classifier $C^b(x)$ with a decision boundary $C^b(x) = 0$ in $\widetilde{X}^b$.

(2) Combine classifiers $C^b(x), \; b = 1, 2, \ldots, B$ by simple majority voting to obtain a final decision rule:

$$\beta(x) = \underset{y \in [-1,1]}{\mathrm{armax}} \sum_b \delta_{sgn}\big(C^b(x)\big).y \tag{6}$$

where $\delta_{ij}$ is the Kronecker symbol and $y \in [-1, 1]$ is a decision (class label) of the classifier.

Reduced-Error Pruning Tree (REPTree)

Quinlan [114] introduced a method based on information gain or variance to build a decision tree that uses reduce-error pruning with back overfitting. The REPTree algorithm sorts values for numerical attributes once; missing values are created using an embedded method by C4.5 in fractional instances.

2.2.4. Comparison and Validation of Gully Erosion Models and Susceptibility Maps

In this section, we introduce the evaluation metrics used in this study. We selected the most widely used metrics based on the machine learning literature, which include machine learning performance evaluation metrics and error metrics.

Machine Learning Evaluation Metrics

Machine learning evaluation metrics include true positive (TP), false positive (FP), precision, recall, F-measure, Matthews correlation coefficient (MCC), receiver operatic characteristic curve (ROC), and the precision recall (PRC) metric. All these metrics are obtained based on the four possibilities shown in Table 1: true positive (TP), false positive (FP), true negative (TN), and false negative (FN). TP and TN are the number of gully erosion pixels that correctly classified as, respectively, gully erosion and non-gully erosion pixels. In contrast, FP and FN pixels are incorrectly classified as gully erosion and non-gully erosion pixels, respectively [76]. The above-monitored metrics can be formulated as follows:

$$\mathrm{Precision} = \frac{\mathrm{TP}}{\mathrm{TP} + \mathrm{FP}} \tag{7}$$

$$\mathrm{Recall} = \frac{\mathrm{TP}}{\mathrm{TP} + \mathrm{FN}} \tag{8}$$

$$F_1 - \mathrm{measure} = 2 \times \frac{(\mathrm{Precision} \times \mathrm{Recall})}{(\mathrm{Precision} + \mathrm{Recall})} \tag{9}$$

We used the Matthews correlation coefficient (MCC) [114] to check the quality of binary (two-class) classifications. This metric has a range from -1 (total disagreement between prediction and observation values) and +1 (perfect prediction). The MCC can be computed as

$$\mathrm{MCC} = \frac{(\mathrm{TP} \times \mathrm{TN}) - (\mathrm{FP} \times \mathrm{FN})}{\sqrt{((\mathrm{TP} + \mathrm{FP})(\mathrm{TP} + \mathrm{FN})(\mathrm{TN} + \mathrm{FP}) - (\mathrm{TN} + \mathrm{FN})}} \tag{10}$$

The receiver operatic characteristic curve (ROC) is a popular and important metric to check the general performance of a model [115]. Recall and 1-specifcty (FP / (FP + TN)) are plotted, respectively, on the *x* and *y*-axes of the ROC. A model with random performance has a straight diagonal line from (0, 0) to (1, 1) on the plot, which thus serves as a reference line. The area under the ROC curve (AUC) is a quantitative measure of the performance of the model. It ranges from 0 (inaccurate model) to 1 (perfect model) [21,116]. The PRC metric is a graph that provides a prediction of future classification performance [117]. The *x*- and *y*-axes are, respectively, recall and precision metrics. The higher the PRC line value, the better the performance of the model.

**Table 1.** Confusion matrix of machine learning models in this study.

|  |  | Predicted Target | |
|---|---|---|---|
|  |  | Gully Erosion (+) | Non-Gully Erosion (−) |
| Actual target | Gully erosion (+) | TP | FP |
|  | Non-gully erosion (−) | FN | TN |

Error-Based Evaluation Metrics

Error-based indexes are the second group of evaluation metrics used to check the performance of the gully erosion mapping. They include Kappa (K), root mean square error (RMSE), relative standard error of the prediction (PRSE), mean absolute error (MAE), and relative absolute error (RAE), which are formulated as

$$\text{Kappaindex (K)} = \frac{A - B}{1 - B} \tag{11}$$

$$A = (TP + TN)/(TP + TN + FN + FP) \tag{12}$$

$$B = ((TP + TN)(TP + FP) + (FP + TN)(FN + TN)/\sqrt{(TP + TN + FN + FP)}) \tag{13}$$

$$\text{RMSE} = \sqrt{\frac{\sum\limits_{i=1}^{n}(p_i - a_i)^2}{n}} \tag{14}$$

$$\text{PRSE} = \frac{\sum\limits_{i=1}^{n}(p_i - a_i)^2}{\sum\limits_{i=1}^{n}(\bar{a} - a_i)^2} \tag{15}$$

$$\text{MAE} = \frac{\sum\limits_{i=1}^{n}|p_i - a_i|}{n} \tag{16}$$

$$\text{RAE} = \frac{\sum\limits_{i=1}^{n}|p_i - a_i|}{\sum\limits_{i=1}^{n}|\bar{a} - a_i|} \tag{17}$$

### 2.2.5. Factor Ranking and Selection by the Information Gain Ratio Technique

Several techniques for factor ranking and selection have been proposed, but the relative advantages and weaknesses of these techniques are unknown [118]. Factor ranking techniques evaluate the relevance of each factor independently and eliminate factors determined to be irrelevant or redundant. They also search for the subset of factors that offers the largest reduction in dimensionality [118].

In this study, we used the information gain ratio (IGR) method to select and rank the most important factors for gully erosion modeling and susceptibility mapping. The IGR method is applied as follows [119]:

Let $T$ be the total number of tuples in the training dataset; $Tj$ as the total number of positive or negative tuples in the training dataset; $v$ is the total number of classes in the dataset; and S is slope angle, which is one of the gully conditioning factors.

$$GainRatio(Slope) = \frac{Gain(Slope)}{SplitInfo(Slope)} \tag{18}$$

$$\text{where; } SplitInfo(T) = -\sum_{j=1}^{v} \frac{|T_j|}{|T|} \log_2 \left( \frac{|T_j|}{|T|} \right) \tag{19}$$

$$Gain(Slope) = I(p,n) - E(Slope) \tag{20}$$

$$E(Slope) = -\sum_{i=1}^{m} \frac{p_i + n_i}{p + n} I(p_i, n) \tag{21}$$

$$I(p_i, n) = -\frac{p}{p+n} \log_2 \frac{p}{p+n} - \frac{n}{p+n} \log_2 \frac{n}{p+n'} \tag{22}$$

$E(Slope)$ represents the entropy of the slope angle factor in the training dataset, $I(p,n)$ denotes the information needed to satisfy a given training dataset, $p$ is the total number of positive tuples in the training dataset, $n$ is the total number of negative tuples in the training dataset, and $m$ is the number of values for the slope angle factor.

## 3. Results

### 3.1. Correlation between Conditioning Factors and Gully Occurrence Using the Frequency Ratio Method

We used the frequency ratio method to calculate the probabilistic relation between gullies as a dependent variable and conditioning factors as independent variables. Figure 5 presents FR values for the classes of each conditioning factor. In the case of rainfall, the 148–159 mm class has the highest FR value (2.15), followed by the 159–171 mm (0.29) and 171–192 mm (0) classes. The >1000 m distance-from-road class had the highest FR value (1.63), followed by the 200–500 m (1.10), 500–1000 m (0.96), 100–200 m (0.77), and 0–100 m (0.73) classes. In the case of NDVI, the 0.12–0.37 class has the highest FR value (3.48). Bare land areas have the highest FR values in the land-use class (1.21). In the case of drainage density, high FR values are associated high drainage density. The 1.62–2.46 km/km$^2$ class, for example has a value of 4.32. For lithology, the Ekgy has by far the largest FR (4.43), followed by Qft2 (1.02), PCK (0.34), and PCmt2 (0.08). No gullies are present on the other lithologies; therefore, their values are 0. Areas located <500 m from rivers have a FR value of 1.63; the more distant classes have 0 values. In the case of profile curvature, the highest FR value (1.18) is associated with the >0.25 class. Flat areas have a FR value of 1.09, which is higher than the values for convex and concave areas (0.72 and 0.53, respectively). The highest FR value for the slope factor is 1.23 (5–10% class). Values for the 0–5% and 10–20% classes are, respectively, 1.07 and 0.76; the 20–30% and >30% classes are 0. Slopes with an eastern aspect have the highest FR value (1.30), following by slopes with northeastern (1.17), southeastern (1.13), northern and flat (1.05), southwestern (0.94), northwestern (0.88), southern (0.70), and western (0.68) aspects. Finally, all gullies are located in areas with an elevation range of 1800–2000 m a.s.l. (FR = 1.16).

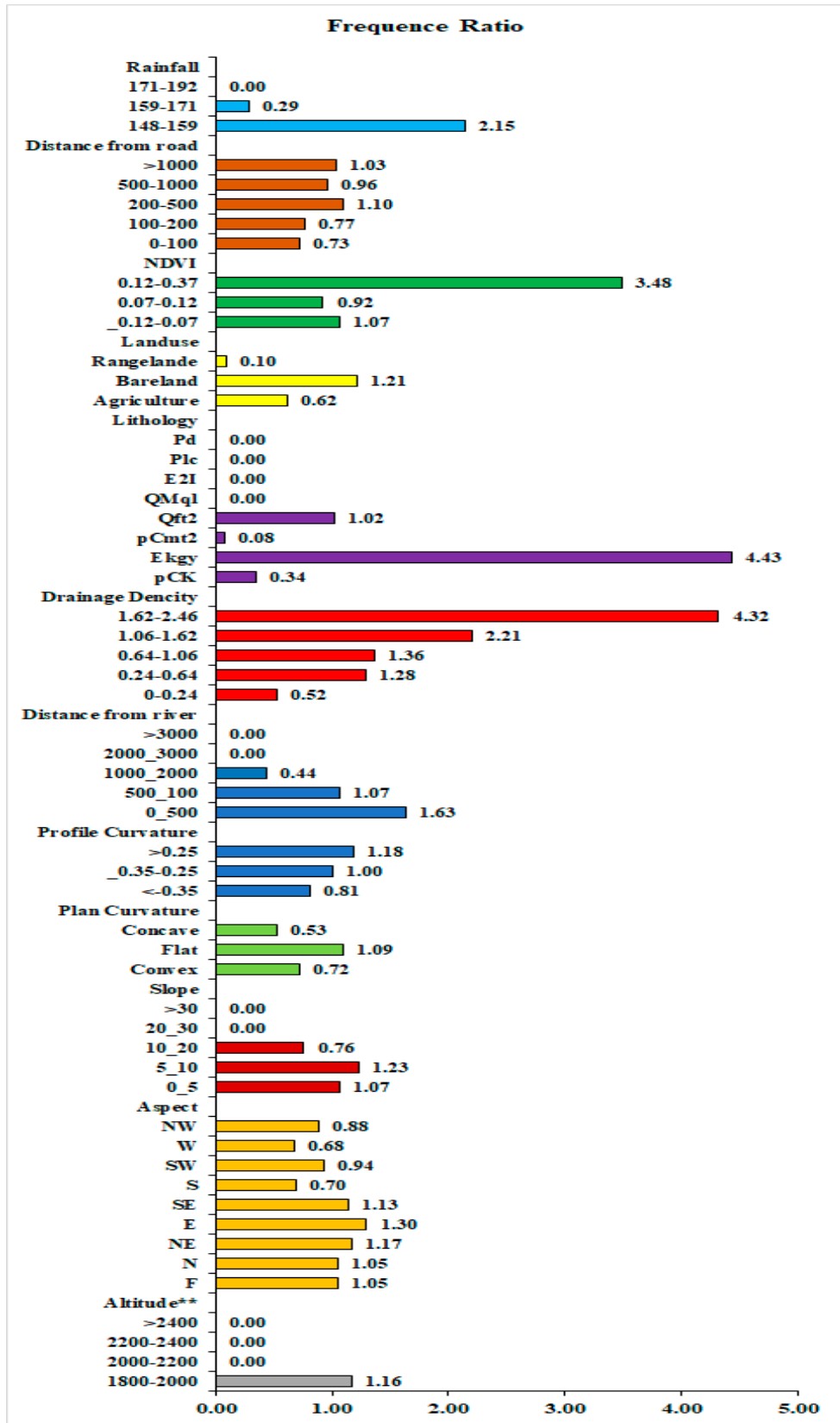

**Figure 5.** Frequency ratios for factors related to gully erosion.

### 3.2. Analysis of Factor Multi-Collinearity

We examined the multi-collinearity of gully erosion conditioning factors using the variance inflation factor (VIF) and tolerances (TOL). Values of VIF >10 and TOL <0.10 generally indicate a multi-collinearity problem [120]. VIF and TOL values for the conditioning factors used in this study are shown in Table 2. The highest VIF and the lowest TOL are, respectively, 2.673 and 0.184, which indicate that there is not a multi-collinearity problem among the conditioning factors and, hence, all factors can be used for gully erosion susceptibility mapping.

**Table 2.** Multi-collinearity statistics for the gully erosion affecting factors.

| Parameters | Collinearity Statistics | |
|:---:|:---:|:---:|
| | Tolerance | VIF |
| Land use | 0.184 | 1.525 |
| Lithology | 0.674 | 1.354 |
| NDVI | 0.628 | 2.047 |
| Plan curvature | 0.492 | 1.254 |
| Profile curvature | 0.398 | 2.673 |
| Rainfall | 0.712 | 1.951 |
| River density | 0.420 | 2.322 |
| River distance | 0.324 | 1.875 |
| Road | 0.583 | 1.840 |
| Slope | 0.809 | 1.245 |
| Aspect | 0.856 | 1.030 |
| Altitude | 0.198 | 2.329 |

### 3.3. The Most Important Factors for Gully Modeling

The average merit (AM) values calculated by information gain ratio (IGR) technique are summarized in Table 3. The results indicate that all factors can be included in gully erosion susceptibility modeling because their AM values are greater than zero. However, rainfall, with an AM value of 0.225, is the most effective factor for gully erosion susceptibility mapping in the study area. It is followed by elevation (AM = 0.186), river density (AM = 0.106), distance to river (AM = 0.093), land use (AM = 0.086), lithology (AM = 0.083), distance to road (AM = 0.038), profile curvature (AM = 0.031), aspect (AM = 0.028), NDVI (AM = 0.023), slope (AM = 0.020), and plan curvature (AM = 0.016).

**Table 3.** The most effective factors for gully erosion occurrence.

| Rank | Conditioning Factor | Average Merit | Standard Deviation |
|------|---------------------|---------------|--------------------|
| 1 | Rainfall | 0.225 | ± 0.012 |
| 2 | Altitude | 0.186 | ± 0.009 |
| 3 | River density | 0.106 | ± 0.011 |
| 4 | River distance | 0.093 | ± 0.015 |
| 5 | Land use | 0.086 | ± 0.007 |
| 6 | Lithology | 0.083 | ± 0.01 |
| 7 | Profile curvature | 0.031 | ± 0.017 |
| 8 | Road | 0.038 | ± 0.014 |
| 9 | Aspect | 0.028 | ± 0.021 |
| 10 | NDVI | 0.023 | ± 0.018 |
| 11 | Slope | 0.02 | ± 0.016 |
| 12 | Plan curvature | 0.016 | ± 0.018 |

### 3.4. Evaluation of Gully Erosion Susceptibility Models

We created four landslide susceptibility models (REPTree, AB-REPTree, Bag-REPTree, and RS-REPTree) using the training dataset. The 10-fold cross-validation method was used to prevent over-fitting and to decrease variability. Heuristic tests were used to find the best values for the parameters of the four models; these are shown in Table 4.

**Table 4.** Parameters of algorithms utilized in this study.

| Methods | Algorithms | Parameters |
|---------|------------|------------|
| Base classifier | Reduced-error pruning tree | Seed, 1; The minimum total weight of the instances in a leaf, 2; Number of folds, 10 |
| Ensembles | Bagging | Seed, 1; The number of iterations, 10 |
| | AdaBoost | Seed, 1; The number of iterations, 10 |
| | Random subspace | Seed, 1; The number of iterations, 10 |

We validated gully erosion susceptibility models using error and machine learning comparison metrics (Tables 5 and 6). The highest values of the Kappa metric were obtained for the RS-REPTree model (0.61), followed by the Bag-REPTree (0.55), AB-REPTree (0.53), and REPTree (0.53) models. The RS-REPTree model has the highest value (0.33) for the MAE metric, followed by the Bag-REPTree (0.28), AB-REPTree (0.24), and REPTree (0.24) models. The RMSE, RAE, and RRSE metrics indicate that the Bag-REPTree model (RMSE = 0.37, RAE = 56.62, and RRSE = 77.75) has the lowest error. It is followed by the RS-REPTree model (RMSE = 0.38, RAE = 67.68, and RRSE = 77.57) and the AB-REPTree model (RMSE = 0.43, RAE = 49.76, and RRSE = 86.49). The REPTree model has the highest error (RMSE = 0.43, RAE = 79.76, and RRSE = 86.50).

**Table 5.** Evaluation of gully erosion susceptibility models using error metrics.

| Models | Kappa | MAE | RMSE | RAE | PRSE |
|--------|-------|-----|------|-----|------|
| REPTree | 0.53 | 0.24 | 0.43 | 79.76 | 86.50 |
| AB-REPTree | 0.53 | 0.24 | 0.43 | 49.76 | 86.49 |
| Bag-REPTree | 0.55 | 0.28 | 0.37 | 56.62 | 75.30 |
| RS-REPTree | 0.61 | 0.33 | 0.38 | 67.68 | 77.57 |

**Table 6.** Evaluation of gully erosion susceptibility models using machine learning metrics.

| Models | TP | FP | Precision | Recall | F-Measure | MCC | AUC | PRSE |
|--------|-----|-----|-----------|--------|-----------|------|------|------|
| REPTree | 0.774 | 0.226 | 0.776 | 0.774 | 0.773 | 0.549 | 0.819 | 0.782 |
| AB-REPTree | 0.768 | 0.232 | 0.77 | 0.768 | 0.767 | 0.537 | 0.844 | 0.838 |
| Bag-REPTree | 0.776 | 0.224 | 0.779 | 0.776 | 0.776 | 0.555 | 0.871 | 0.866 |
| RS-REPTree | 0.806 | 0.194 | 0.809 | 0.806 | 0.805 | 0.615 | 0.874 | 0.865 |

The machine learning comparison metrics shown in Table 6 indicate that the RS-REPTree model performed best based on TP, FP, precision, recall, F-measure, MCC, AUC, and PRSE values. It is followed by the Bag-REPTree, REPTree, and AB-REPTree models in terms of TP, FP, precision, recall, F-measure, and MCC metrics. The AB-REPTree model performed better than the REPTree model in term of the AUC and PRSE metrics.

## 3.5. Development of Gully Erosion Susceptibility Maps

We calculated gully erosion susceptibility indices for each cell based on the results of the ensemble models. We then constructed gully erosion susceptibility maps for the study area using the Ada-REPTree, Bag-REPTree, REPTree, and RS-REPTree models (Figure 6). Gully erosion susceptibility classes (low, moderate, high, and very high) were created using the natural breaks method. For example, in the case of the Ada-REPTree map, the four susceptibility classes have values of 0.00–0.13, 0.13–0.42, 0.42–0.78, and 0.78–1.00 (Figure 6a). Comparison of the four maps indicates that the REPTree model predicts a larger part of the watershed as having high and very high erosion susceptibilities. More generally, the maps show that most cells of low erosion susceptibility are located on steep slopes in the marginal parts of the watershed. The high and very high susceptibility classes cover the northern and central parts of the watershed where most of the observed gully sites are located.

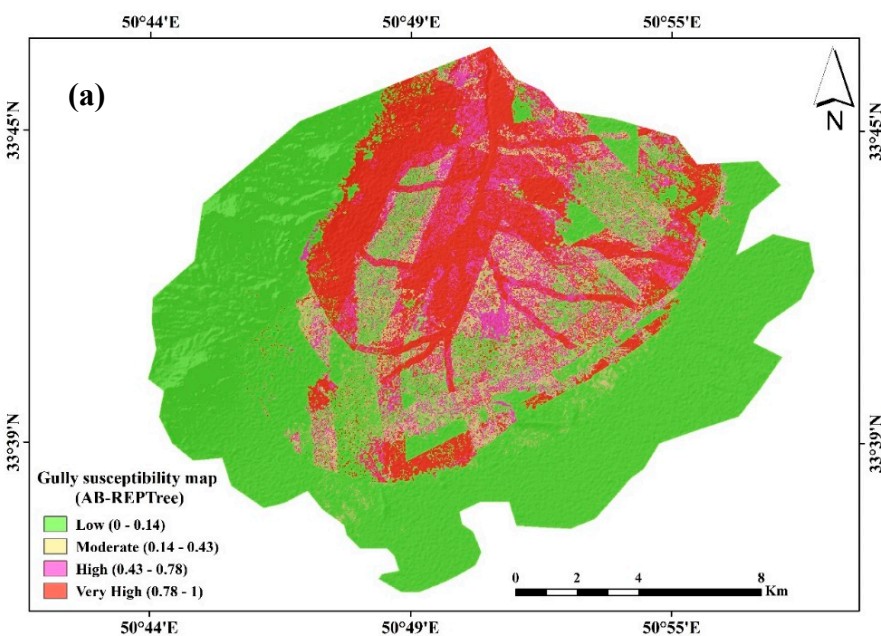

**Figure 6.** *Cont.*

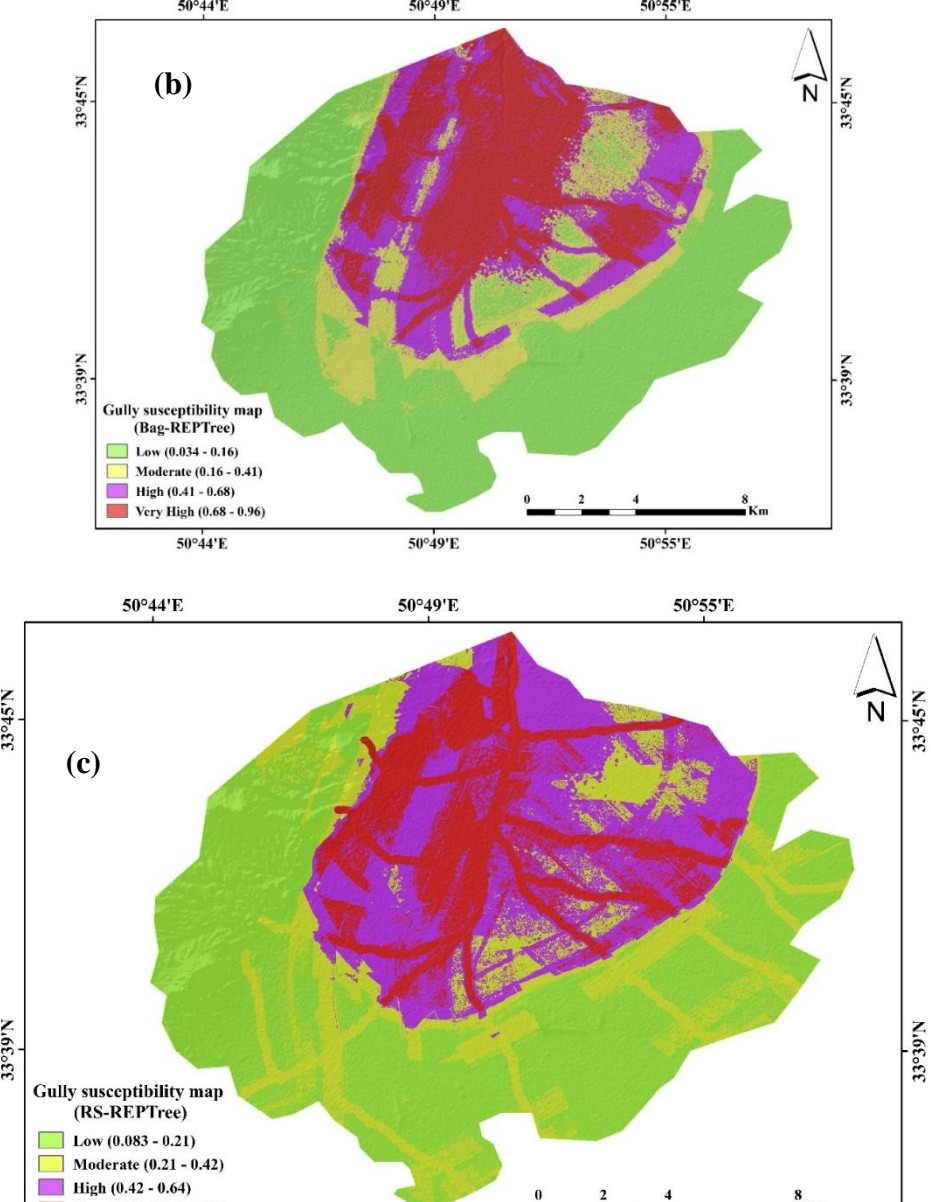

**Figure 6.** *Cont.*

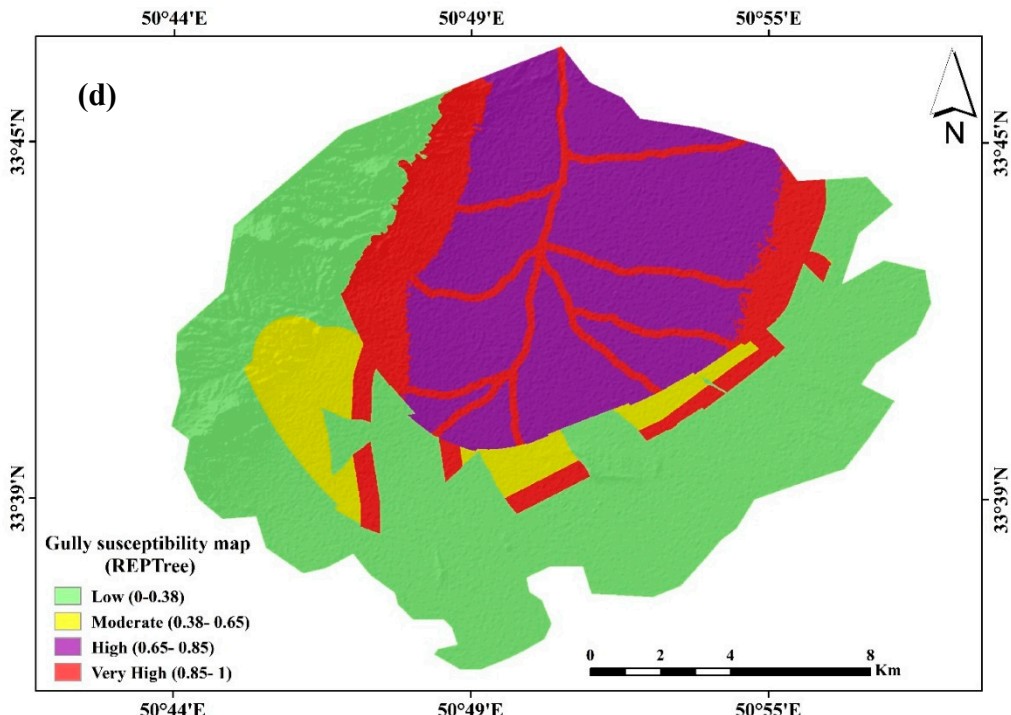

**Figure 6.** Gully erosion susceptibility maps based on: (**a**) AB-REPTree, (**b**) Bag-REPTree, (**c**) RS-REPTree, and (**d**) REPTree.

### 3.6. Evaluation and Comparison of the Models

Evaluation of model performance is an important step in the spatial modeling process [88]. In this study, we evaluated the performance of the four ensemble models using the area under the ROC curve (AUC), standard error (SE), and 95% confidence interval for the training and testing datasets. The logistic regression (LR) model was used as a benchmark method. ROC curves for the training dataset are shown in Figure 7. The curves show that all tested ensemble models perform well in spatially predicting gully erosion susceptibility. However, the ROC curve for the REPTree model falls below the curves of the other models. Other results of the goodness-of-fit analysis of the training dataset are shown in Table 7. These results indicate that RS-REPTree model has the best performance with the highest AUC (0.874), lowest SE value (0.0191), and narrowest 95% CI (0.834–0.907). Sequentially, the Bag-REPTree, AB-REPTree, and REPTree models have slightly lower performances. Finally, the performances of three ensemble models are better than that of the benchmark LR model.

**Table 7.** ROC curve using the training dataset.

| Variable | AUC | SE | 95% CI | |
|---|---|---|---|---|
| | | | Lower Bound | Upper Bound |
| **REPTree** | **0.819** | 0.0238 | 0.774 | 0.859 |
| AB-REPTree | 0.844 | 0.0210 | 0.801 | 0.881 |
| Bag-REPTree | 0.871 | 0.0191 | 0.830 | 0.905 |
| RS-REPTree | 0.874 | 0.0191 | 0.834 | 0.907 |
| LR | 0.825 | 0.0222 | 0.780 | 0.864 |

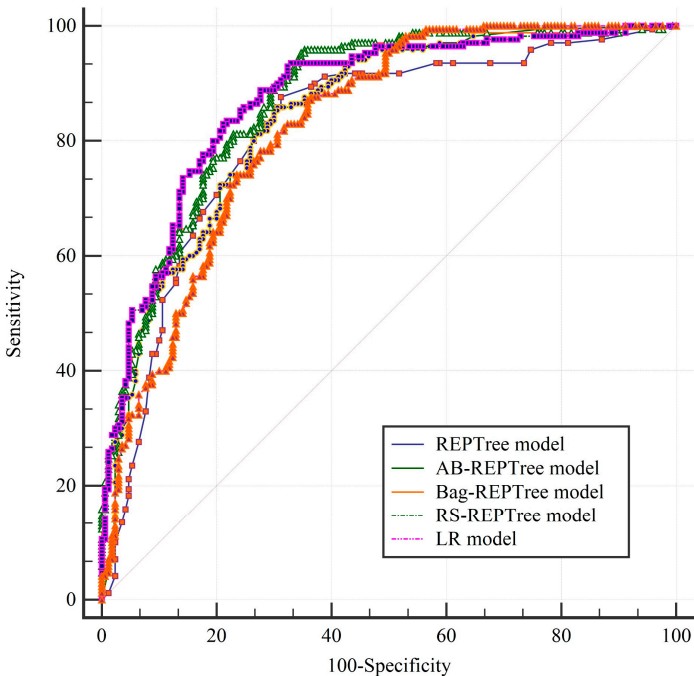

**Figure 7.** ROC curves related to the susceptibility models used in this study.

Model performances for the testing dataset based on the ROC curve, AUC, SE, and 95% CI values are shown in Figure 8 and summarized in Table 8. All models performed well, but the proposed new ensemble model, RS-REPTree, has the highest prediction capability based on its AUC (0.860), SE (0.0315), and 95% CI (0.793–0.912). It is followed by the Bag-REPTree (AUC = 0.841), LR (AUC = 0.824), AB-REPTree (AUC = 0.805), and REPTree (AUC = 0.800) models. Overall, our results show that the new ensemble models of REPTree outperform and outclass the standard REPTree model in gully erosion susceptibility mapping.

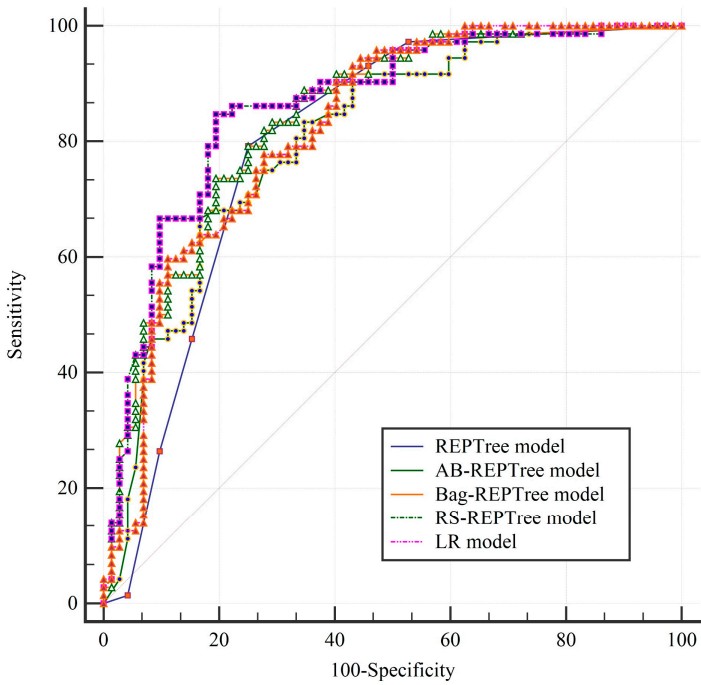

**Figure 8.** ROC curve and AUC of the models: (**a**) training dataset, and (**b**) validation dataset.

**Table 8.** ROC curve using the validation dataset.

| Model | AUC | SE | 95% CI | |
| --- | --- | --- | --- | --- |
| | | | Lower Bound | Upper Bound |
| **REPTree** | **0.800** | 0.0383 | 0.725 | 0.862 |
| AB-REPTree | 0.805 | 0.0368 | 0.731 | 0.866 |
| Bag-REPTree | 0.841 | 0.0329 | 0.771 | 0.896 |
| RS-REPTree | 0.860 | 0.0315 | 0.793 | 0.912 |
| LR | 0.824 | 0.0350 | 0.751 | 0.882 |

## 4. Discussion

Obtaining reliable map of gully erosion susceptibility remains yet a challenge for managers, land use planners, and engineers. To address this challenge, researchers are proposing new models and testing them in different gully-prone regions around the world. In this paper, we propose and evaluate three ensembles of the REPTree model for gully erosion susceptibility mapping. The modeling process is based on an investigation of the relationships between spatial locations of gullies in the Rabat Turk watershed and a suite of different geo-environmental factors. We demonstrate that rainfall, elevation, river density, distance to rivers, land-use, and lithology are important factors for gully erosion in the study area. In contrast, plan curvature, slope, NDVI, aspect, and distance to roads are the less important.

An examination of the literature suggests that conditioning factors for gully erosion are area-specific and cannot be reliably extrapolated to other regions. For example, Amiri et al. [121] identified land-use as the most important factor in their study areas, whereas Rahmati et al. [122] and Garosi et al. [92] reported that distance from rivers is the most important factor in their studies. Furthermore, the slope factor, which we and Rahmati et al. [122] ranked as a relatively unimportant factor, was among the most effective factors identified by Rahmati et al. [97]. These differences call for further research on controls of gully erosion in different landscapes.

The ensemble learning techniques used in this study (AB, bagging, and RS) improved the goodness-of-fit and prediction performance of REPTree. Among these techniques, random subspace outperformed the other two techniques in improving both the training and validation of the base REPTree model. The RS ensemble learning technique performed better than the other techniques in decreasing the variance, bias, and noise of the modeling process, and protected the models from over-fitting. The superiority of the RS ensemble learning technique stems from the use of random subspaces for aggregating the base classifiers, which results in better performance compared to the original feature space [112]. In addition, the base classifier works better using smaller subspaces, as shown by Pham et al. [123]. The literature includes numerous successful applications of RS ensemble learning techniques for predicting different types of natural hazards. For example, Tien Bui et al. [76] showed that the naive Bayes tree performed better when used in combination with the RS technique for landslide modeling, and Shirzadi et al. [62] demonstrated that the RS technique improved the performance of the alternating decision tree base classifier.

Our results suggest that the Bagging technique is the second-best ensemble learning method for improving REPTree performance, which is in line with previous findings. For example, Hong et al. [124] reported that bagging, used in combination with the j48 decision tree, has higher predictive capacity than the single j48 and AB-j48 models alone. In another study, Bui et al. [58] reported that the functional tree (FT) model with bagging outperforming the AB-FT method.

Although our study is the first to use REPTree in combination with ensemble learning techniques for gully erosion modeling, this approach has been used by Pham et al. [123] for predicting landslides. They ranked the ensemble models in terms of prediction capability, from best to worst, to be: BA-REPTree (AUC = 0.872), rotation forest REPTree (AUC = 0.872), RSRETree (AUC = 0.864), and MultiBoost REPTree (AUC = 0.855). The differences in their results and ours suggest that the techniques are case-

and site-specific and that their performances depend heavily on the datasets that are trained and built upon.

Although it is difficult to directly compare the results of this study with those reported from other regions, we suggest that our ensemble models perform better than the generalized linear model (AUC = 0.71), boosted regression tree (AUC = 0.84), multivariate adaptive regression spline (AUC = 0.83), and ANN (AUC = 0.84) models used by Garosi et al. [104]; the certainty factor model (AUC = 0.82) used by Azareh et al. [82]; and the Fisher's linear discriminant analysis (AUC = 0.76), logistic model tree (AUC = 0.77), and NBT (AUC = 0.78) models of Arabameri et al. [125]. In contrast, however, our models were outperformed by the maximum entropy (AUC = 0.88, 0.90) models used by Azareh et al. (2019) and Kariminejad et al. [107]; BFTree and its ensembles (bagging and RS) (AUC = 0.92) used by Hosseinalizadeh et al. [81]; and the multivariate additive regression splines (AUC = 0.91), SVM (AUC = 0.88), and FR (AUC = 0.96) models employed by Gayen et al. [126]. Again, these different results are attributable to local differences in the environments in which the models were used.

Our field survey indicated that gullies in the study area are located along tributaries near the main river in the Rabat Turk study area. Erosion is initiated by focusing of runoff along these tributaries, gradual gully retrogression, and piping above gully heads. Gullies on the east side of the river have lower slopes than those on the west side of the river, perhaps because there is little vegetation in the former areas. There is also more upslope area for gully development on the west side of the river, allowing for more flow with the gully system. Our results are in agreement with the findings of Vandekerckhove et al. [127] and Bergonse and Reis [128], who argued that gullies are mainly formed through extreme runoff related to slope-area relations. The gully erosion susceptibility map of the study area obtained using the RS-REPTree ensemble model accurately predicts observed gullies along the main river and its tributaries.

Despite the improved prediction performance provided by ensemble models, the difficulty associated with proper parameter tuning still restricts their development and application. In this study, we manually tuned the parameters of the ensemble methods through a trial-and-error process [129,130]. There are, however, several optimization techniques (e.g., metaheuristic optimization algorithms) that can significantly speed up the process of model building [131,132]. Nevertheless, ensemble models are easy to develop within open-source WEKA software and do not require advanced programming knowledge. They can be applied to types of environmental research that involve datasets with a number of geo-environmental variables and a set of presence/absence locations of the phenomenon being modeled. Such datasets can be generated with automated GIS techniques from accessible geospatial data (e.g., DEM, soil, lithology, and meteorological records).

## 5. Conclusions

Gully erosion is an advanced stage of water erosion and sediment production that can transfer large volumes of sediment into stream channels, resulting in environmental damage. It is a common problem in arid and semi-arid landscapes, and therefore, prediction and mapping of areas susceptible to gully erosion are of interest to soil scientists, natural resource authorities, and land managers. Accordingly, researchers have used a variety of machine learning methods to understand the causes of gully erosion and to produce reliable erosion susceptibility maps [133].

We addressed this problem by studying gully erosion in a sub-basin of the Shoor River watershed in Isfahan Province (Iran), which has a semi-arid climate and a human-impacted landscape. We used 12 conditioning factors tested by the information gain ratio method, and REPTree coupled with the AB, BA, and RS ensemble learning methods to model gully erosion and produce gully erosion susceptibility maps. The following are key conclusions of our study:

(1) Rainfall, elevation, and river density are the most important factors for gully erosion in the study area. Most gully erosion sites are located in areas of lower rainfall and lower elevation.

(2) REPTree and all its ensembles yielded a high goodness-of-fit and prediction accuracy during the modeling process, but the ensemble RS-REPTree performed best. RS decreased over-fitting and noise in the training datasets, which resulted in better prediction. It successfully predicted gully erosion locations and allowed us to produce an accurate gully erosion susceptibility map of the study area.

(3) Modeling gully erosion is a complicated task, with many uncertainties. The proposed machine learning model is an easy-to-use, inexpensive decision-making tool that can supplement expensive field surveys. It also provides managers with guidance on what further information might be needed to provide a more accurate map of gully erosion.

(4) Gully erosion susceptibility maps are essential products for hazard analysis and management. We recommend our proposed ensemble RS-REPTree model for predicting gully erosion in other semi-arid and arid areas. However, the performance of this model depends on the quality of the data used.

(5) We recommend further research on other hybrid data mining methods, as well as ensemble boosting algorithms with REPTree. We also recommend further sensitivity analysis of gully erosion conditioning factors.

**Author Contributions:** V.-H.N., S.J., M.A., W.C., M.F., E.O., A.S., H.S., J.J.C., A.J., F.M., B.T.P., B.B.A., and S.L. contributed equally to the work. S.J., M.A., M.F., and F.M. collected field data and conducted the gully erosion mapping and analysis. S.J., M.A., W.C., M.F., E.O., A.S., H.S., A.J., and F.M. wrote the manuscript. V.-H.N., A.S., H.S., J.J.C., B.T.P., B.B.A., and S.L. provided critical comments in planning this paper and edited the manuscript. All the authors discussed the results and edited the manuscript. All authors have read and agreed to the published version of the manuscript.

**Funding:** This research was supported by the Basic Research Project of the Korea Institute of Geoscience, Mineral Resources (KIGAM) funded by the Minister of Science and ICT.

**Conflicts of Interest:** The authors declare no conflict of interest.

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
