# Peer review of "GIS-Based Gully Erosion Susceptibility Mapping: A Comparison of Computational Ensemble Data Mining Models"

_applsci, doi:10.3390/app10062039_

Round 1

Reviewer 1 Report

The title of the manuscript (MS) deals with GIS-based gully erosion susceptibility mapping: A comparison of computational ensemble data mining models. The MS is globally sound and well written, however, it needs a bit more work to be considered for publication.
My main concern is about the Abstract, Discussions, and Conclusion. I have a number of relatively easily addressed comments detailed below.

Specific Comments:

In the "Abstract" of the MS, the objectives should be presented briefly and concisely. Well defined research objectives will help the readers to clearly understand the study. In addition, finding results (values) should be added to the Abstract.

In the "Materials and Methods" section, you have some typos in the equations, see eq.9. (F - measure).

In the "Discussions" section, the authors must extend the comparison between their approach and other ones that have been developed and used in the literature for the same or related purposes (I recommend increasing the number of Scientific articles cited, especially to compare the study context with similar studies). Also, in this section, the authors should also highlight the current limitations and usefulness of the proposed research, and briefly mention some precise directions that they intend to follow in their future research work.
Most importantly in the "Discussion" section, the authors should talk about the applicability of this method to another case study abroad. What are the mandatory prerequisites? Is it applicable to worldwide?

In conclusion, describe how a previously identified gap in the literature has been filled by your research (demonstrate the importance of your idea). Elaborate on the impact and significance of your findings. Finally, introduce possible new or expanded ways of thinking about the research problem based on the results. Mention the applicability of this method to another case study abroad.

Author Response

Dear Editor

We greatly appreciate the meticulous review of our research article. We would like to thank the reviewers for their insightful and constructive comments and suggestions regarding several aspects of the manuscript. We have used them to our advantage and improved the manuscript. As described in detail below, we have addressed the concerns raised by the reviewers and made modifications to the revised manuscript as suggested. We have shown our response below each comment in blue.The changes are highlighted in the revised manuscript by Red color. We hope our revision has improved the paper to a level of your satisfaction.

Authors

Responses to Reviewers Comments

Reviewer # 1

The title of the manuscript (MS) deals with GIS-based gully erosion susceptibility mapping: A comparison of computational ensemble data mining models. The MS is globally sound and well written, however, it needs a bit more work to be considered for publication. My main concern is about the Abstract, Discussions, and Conclusion. I have a number of relatively easily addressed comments detailed below.

Response:

Thank you for the valuable and constructive commnet. We understand esteemed reviewr concernes about abstract, discussion and conclusion and we agree with her/his opinions. Therefor, we check again theses sectiones and they were carfully read in whcich we added some more imporstant infprmations to them. Plaese refer to the revsied version of the manuscript. 

Specific Comments:

In the "Abstract" of the MS, the objectives should be presented briefly and concisely. Well defined research objectives will help the readers to clearly understand the study. In addition, finding results (values) should be added to the Abstract.

Response:

Thank you for your good comment. We agree with you that both objectives and inally results should be more enhanced by more information. Accordinlgy, we added some information to the abstract of revised version of the manuscript. However, we copy here for fast revision by esteemed reviewer:

“The main objective of this research is to accurately detect and predict the gully erosion prone areas to reduce costly preventive practices from gully erosion developing. To do that, in this paper, we couple hybrid models of a commonly used base classifier (reduced pruning error tree, REPTree) with Adaboost (AB), Bagging (Bag), and Random Subspace (RS) algorithms to created gully erosion susceptibility maps and also comparing their performances to model the occurrence of gully erosion in a sub-basin of the Shoor River watershed in northwestern Iran.”

“Although, all three hybrid models that we tested significantly enhanced and improved the predictive power of REPTree (AUC=0.800), the RS-REPTree (AUC= 0.860) model outperformed and outclassed the Bag- REPTree (AUC= 0.841), and the AB- REPTree (AUC= 0.805) models. Therefore, the RS-REPTree hybrid proposed model can be used as an alternative tool by decision makers, planers and environmental engineers to better management of gully erosion prone areas.” 

In the "Materials and Methods" section, you have some typos in the equations, see eq.9. (F – measure)

Response:

Thank you for your attention. We agree with you and we chekced line by line whole the manuscript and the error typos, such as equation 9, were known and detected.

In the "Discussions" section, the authors must extend the comparison between their approach and other ones that have been developed and used in the literature for the same or related purposes (I recommend increasing the number of scientific articles cited, especially to compare the study context with similar studies). Also, in this section, the authors should also highlight the current limitations and usefulness of the proposed research, and briefly mention some precise directions that they intend to follow in their future research work. Most importantly in the "Discussion" section, the authors should talk about the applicability of this method to another case study abroad. What are the mandatory prerequisites? Is it applicable to worldwide?

Response:

Thank you for the good comment. In the revised version of the manuscript, we have improved the discussion section to address the reviewer’s suggestion. We have now broadly compared our findings with those reported in the literature, expressed limitations and usefulness of the proposed models, and emphasized the applicability of this methodology to other regions around the world with their mandatory prerequisites. We copy here for fast revisin:

Although it is difficult to directly compare these current results with those reported in previous works, since they have often been reported from different regions, with this difference in mind, the ensemble models presented here performed better than the generalized linear model (AUC 0.71), boosted regression tree (AUC 0.84), multivariate adaptive regression spline (AUC 0.83), and ANN (AUC 0.84) models used by Garosi et al. [118], the certainty factor model (AUC = 0.82) by Azareh et al. [82], the Fisher’s linear discriminant analysis (AUC = 0.76), logistic model tree (AUC = 0.77), and NBT (AUC = 0.78) models by Arabameri et al. [119]. However, our models were outperformed by the maximum entropy (AUC = 0.88, 0.90) from Azareh et al. (2019) and Kariminejad et al. [100], the BFTree and its ensembles (Bag and RS) (AUC 0.92) by Hosseinalizadeh et al. [81], and the multivariate additive regression splines (AUC = 0.91), SVM (AUC = 0.88), and FR (AUC = 0.96) from Gayen et al. [120]. Again, these different results can be attributed to local conditions of the landscapes that caused different model performance.

Despite the improved prediction performance using these ensemble models, the difficulty associated with proper parameter tuning may restrict the development and application of such models. In this study, we manually tuned the parameters of the ensemble techniques via a trial and error process [121, 122], whereas there are several optimization techniques (e.g., metaheuristic optimization algorithms) that can significantly speedup the process of model building [123-125]. Ignoring this drawback, these models are quite easy to use to develop within the open source WEKA software, and do not require profound programming knowledge. The applicability domain of these models can be extended to all types of environmental research on which the dataset of the models has been generated. The dataset includes a number of geo-environmental variables and a set of presence/absence location of the phenomenon being modelled. Such a dataset can be easily generated via automated GIS techniques using broadly accessible geospatial data (e.g., DEM, soil, lithology, and meteorological records).

In conclusion, describe how a previously identified gap in the literature has been filled by your research (demonstrate the importance of your idea). Elaborate on the impact and significance of your findings. Finally, introduce possible new or expanded ways of thinking about the research problem based on the results. Mention the applicability of this method to another case study abroad.

Response:

Thank you for the good opinion. We agree with you and based on your suggestion the conclusion section were revised and added some information to enahnce the quality of this section. We copy here for fast revision:

 “A number of approaches have been proposed and studied by a number of literature researchers. But every method has weak points that need to be addressed. The need for a precise and reliable method to identify areas affected by gully erosion has led the researcher to use ensemble of different methods to enhance the ability of individual methods.”

“Modeling the hazard of gully erosion is a complicated and multitude of uncertainties. The proposed machine learning model provides a decision-making method for assessing the hazards and hazards of gully erosion compared to field surveys of pricier and less expensive methods. It also provides managers with guidance on what further information may be needed to provide a more accurate map of the gully erosion in order to reduce further damage. Gully erosion susceptibility maps are essential products for further analysis and disaster management and risk mapping. Therefore, we recommend the proposed RS-REPTree model for mapping and predicting gully erosion in other semi-arid and arid areas. However, the predictability of this model depends on the quality of the data used.  Furthermore, researching other hybrid data mining methods as well as ensemble boosting algorithms with REPTree and performing sensitivity analysis of conditioning factors are great topics for future studies that will probably lead to more consistent results.”

Reviewer 2 Report

Overall the manuscript is well laid out and clearly states the overall objective and well as the results are well presented. From the manuscript it looks like the size of the data is small and using data intensive ensembling approaches might lead to overfitting. It would be interesting to see the distribution of the estimation and testing data.

It is not clear to me what the size of the data set it. It says there were 242 gully locations and 242 non-gully locations. Assuming the total data point is 482 (70% was used for training). Given the size of the data, I am not sure if the size of the data set is large enough foe tree-based algorithms. It would be good to use logistic regression as the benchmark and then compare it to other ensemble algorithms.

It is not clear what were the hyperparameters chosen for the RPTree algorithms. Usually, the practice is to estimate the hyperparameters based on the data using nested sampling, where the outer cross validation is used to tune the parameters and the inner cross-validation is used to test the parameters on the unseen data, this avoids any data leakage.

Another question is since tree-based algorithms tend to overfit and small sample exacerbates the problem. In suck cases it would be good to show the model stability by performing multiple holdout testing and reporting the average across holdouts.

What I would like to see are the following

1)      Size and distribution of the estimation data.

2)      How much of the performance increase is obtained by the ensemble algorithms over logistic regression.

3)      Choice of hypermeters for the ensemble algorithms.

4)      Stability of the proposed methods.

Author Response

Reviwer 2

We greatly appreciate the meticulous review of our research article. We would like to thank the reviewers for their insightful and constructive comments and suggestions regarding several aspects of the manuscript. We have used them to our advantage and improved the manuscript. As described in detail below, we have addressed the concerns raised by the reviewers and made modifications to the revised manuscript as suggested. We have shown our response below each comment in blue.The changes are highlighted in the revised manuscript by Red color. We hope our revision has improved the paper to a level of your satisfaction.

Authors

Responses to Reviewers Comments

Reviewer # 2

Overall the manuscript is well laid out and clearly states the overall objective and well as the results are well presented. From the manuscript it looks like the size of the data is small and using data intensive ensembling approaches might lead to overfitting. It would be interesting to see the distribution of the estimation and testing data.

Response:

Thank you for your attention. Although the number of dataset is low and over-fitting maybe occur; however, we selects training and testing dataset using trial and error technique about 20 times to overcome this problem and achieve a training dataset by the highest goodness-of-fit and prediction accuracy.  Figure 1 show the distribution of training and testing dataset of the study area.  

It is not clear to me what the size of the data set it. It says there were 242 gully locations and 242 non-gully locations. Assuming the total data point is 482 (70% was used for training). Given the size of the data, I am not sure if the size of the data set is large enough foe tree-based algorithms. It would be good to use logistic regression as the benchmark and then compare it to other ensemble algorithms.

Response:

Thank you for the comment. We in this study had 242 gully locations over the study area that they were considered as gully occurrence as “1” code. As you know, in machine learning classification techniques the dependent variable should be binary and thus we randomly selected 242 non-gully locations as non-gully occurrence as “0” code. Therefore, we tests a ratio of 70/30 about 20 times and finally we find the best combination of gully and non-gully location with the highest goodness-of-fit and prediction accuracy that are shown in Figure 1. According to the best of our knowledge of literature review, the number of training dataset is not more important for modeling process because the over-fitting can be decreased by some techniques for example changing in combination of samples (e.g., ratio of 70/30 and 80/20) and also using of 10-fold cores validation technique. However, the distribution of training landslide locations on conditioning factors and selecting a proper algorithm are a crucial role to obtain the best goodness of fit (using training dataset) and prediction accuracy (using validation dataset) in the modelling and evaluation process. Moreover, there are some references that with lower training dataset of our training dataset have been achieved the reliable and reasonable results (Yalcin, 2008; Jaafari et al., 2014; Shahabi et al., 2015; Bui et al., 2016; Pham et al., 2016).

We agree with you to run the logistic regression (LR) model to check the general performance of the new hybrid proposed model. We based on your suggestion performed the LR and the comparison results were added to the revised version of the manuscript. We copy here some valuable Tables and Figures for fast revision:

Yalcin, A (2008) In their study for comparing landslide susceptibility map prepared using analytical hierarchy process and bivariate statistics in Ardesen, Turkey, only used of 40 landslide locations for modelling and prediction. Jaafari et al (2014) applied of 103 landslide locations to prepare landslide susceptibility maps by frequency ratio and index of entropy models. 72 cases (70 %) out of 103 detected landslides were randomly selected for modeling (training), and the remaining 31 (30 %) cases were used for the model validation purposes. Shahabi et al (2015) using a total of 92 landslide locations, the training and validation process (80 % (74 cases) and 20 % (18 cases)) have been performed based on the WLC, AHP and SMCE models. Result of prediction capability using validation datasets showed that the SMCE (AUC=0.937) had the highest power prediction followed by AHP (AUC=0.871) and WLC (AUC=0.841), respectively. Bui et al (2016) for flood susceptibility mapping have been used of a total of 74 flood locations with a ratio of 70/30 for training (54 locations) and validating (22 locations) models, respectively. Experimental results based on neural fuzzy inference system and metaheuristic optimization for flood susceptibility modeling, namely MONF showed that the proposed model has high performance on both the training (RMSE = 0.306, MAE = 0.094, AUC = 0.962) and validation dataset (RMSE = 0.362, MAE = 0.130, AUC = 0.911). Pham et al (2016) have been used of only 65 landslide locations in their study. They partitioned these locations into 70 % (45 locations) and 30 % (20 locations) for training and validation dataset, respectively. Results concluded that the area under the ROC curve for the LSSVM and MADT models were 0.803 and 0.853, respectively.

“3.6. Evaluation and comparison of the models

Evaluation of model performance is a major step in the spatial modeling process [88]. In this study, we evaluated the performance of the four developed ensemble models using area under the ROC curve (AUC), standard error (SE), and 95% confidence interval for the training and test datasets. Besides, the logistic regression (LR) model was used as benchmark method. ROC curves for the training dataset are shown in Figure 7. The curves show that all tested ensemble models perform well in spatially predicting gully erosion susceptibility. However, the ROC curve for the REPTree model falls below the curves of the other models. Other results of the goodness-of-fit analysis for training dataset are shown in the Table 6. These results indicate that RS-REPTree model has the best performance with the highest AUC (0.874), lowest SE value (0.0191), and narrowest 95% CI (0.834-0.907). In succession, the Bag-REPTree, AB-REPTree, and REPTree models have slightly lower performances. In addition, the performances of three ensemble models’ are better than that of the benchmark LR model.

Figure 7. ROC curves related to the susceptibility models used in this study.

Table 6. ROC curve using the training dataset.

Variable

AUC

SE

95% CI

Lower bound

Upper bound

REPTree

0.819

0.0238

0.774

0.859

AB-REPTree

0.844

0.0210

0.801

0.881

Bag-REPTree

0.871

0.0191

0.830

0.905

RS-REPTree

0.874

0.0191

0.834

0.907

LR

0.825

0.0222

0.780

0.864

The performances of the investigated models for the test dataset based on the ROC curve, AUC, SE, and 95% CI values are shown in Figure 8 and summarized in Table 7. All models performed well, but the proposed new ensemble model, RS-REPTree, has the highest prediction capability based on AUC (0.860), SE (0.0315), and 95% CI (0.793-0.912). It is followed by the Bag-REPTree (AUC = 0.841), LR (AUC=0.824), AB-REPTree (AUC = 0.805), and REPTree (AUC = 0.800) models. Overall, our results show that the new ensemble models of REPTree outperform and outclass the standard REPTree model in gully erosion susceptibility mapping.

Figure 8. ROC curve and AUC of the models: (a) training dataset, and (b) validation dataset

Table 7. ROC curve using the validation dataset

Model

AUC

SE

95% CI

Lower bound

Upper bound

REPTree

0.800

0.0383

0.725

0.862

AB-REPTree

0.805

0.0368

0.731

0.866

Bag-REPTree

0.841

0.0329

0.771

0.896

RS-REPTree

0.860

0.0315

0.793

0.912

LR

0.824

0.0350

0.751

0.882

It is not clear what were the hyperparameters chosen for the REPTree algorithms. Usually, the practice is to estimate the hyperparameters based on the data using nested sampling, where the outer cross validation is used to tune the parameters and the inner cross-validation is used to test the parameters on the unseen data, this avoids any data leakage.

Response:

Thank you for your attention. We agree with you that the modeling process by the ensemble models need to be clearly by identifying the hyperparameters of the models. Therefore, we added a table that can address the esteemed reviewer concern. We copy here for fast revision:

“Heuristic tests were used to find the best values for the four model’s parameters, which are shown in Table 3".”

Table 3 Parameters of algorithms utilized in this study.

Methods

Algorithms

Parameters

Base classifier

Reduced-error pruning tree

Seed, 1ï¼›The minimum total weight of the instances in a leaf, 2; Number of folds, 10

Ensembles

Bagging

Seed, 1; The number of iterations, 10

AdaBoost

Seed, 1; The number of iterations, 10

Random Subspace

Seed, 1; The number of iterations, 10

Another question is since tree-based algorithms tend to overfit and small sample exacerbates the problem. In such cases it would be good to show the model stability by performing multiple holdout testing and reporting the average across holdouts.

Response:

Thank you for your attention. As we mentioned the earlier comment, we selects training and testing datasets using trial and error technique about 20 times to overcome this problem and achieve a training dataset by the highest goodness-of-fit and prediction accuracy. Additionally, we used of 10-fold cross-validation method to avoid over-fitting and decrease the variability.

What I would like to see are the following

Size and distribution of the estimation data.

Response:

We in this study had 242 gully locations that were classified into a ratio of 70%, 169 gullies, as training and 30%, 73 gullies, as testing datasets. Additionally, we selected 242 non-gully locations that it was classified also into a ratio of 70%, 169 non-gullies, as training and 30%, 73 non-gullies, as testing datasets.. Therefore, overall we had 338 gully and non-gully locations for training dataset and also 146 gully and non-gully locations for testing dataset. 

How much of the performance increase is obtained by the ensemble algorithms over logistic regression.

Response:

Thank you for the comment. As we mentioned in the earlier comment, we aimed in this study to compare the best hybrid model based on REPTree decision tree algorithms.

“3.6. Evaluation and comparison of the models

Evaluation of model performance is a major step in the spatial modeling process [88]. In this study, we evaluated the performance of the four developed ensemble models using area under the ROC curve (AUC), standard error (SE), and 95% confidence interval for the training and test datasets. Besides, the logistic regression (LR) model was used as benchmark method. ROC curves for the training dataset are shown in Figure 7. The curves show that all tested ensemble models perform well in spatially predicting gully erosion susceptibility. However, the ROC curve for the REPTree model falls below the curves of the other models. Other results of the goodness-of-fit analysis for training dataset are shown in the Table 6. These results indicate that RS-REPTree model has the best performance with the highest AUC (0.874), lowest SE value (0.0191), and narrowest 95% CI (0.834-0.907). In succession, the Bag-REPTree, AB-REPTree, and REPTree models have slightly lower performances. In addition, the performances of three ensemble models’ are better than that of the benchmark LR model.

Figure 7. ROC curves related to the susceptibility models used in this study.

Table 6. ROC curve using the training dataset.

Variable

AUC

SE

95% CI

Lower bound

Upper bound

REPTree

0.819

0.0238

0.774

0.859

AB-REPTree

0.844

0.0210

0.801

0.881

Bag-REPTree

0.871

0.0191

0.830

0.905

RS-REPTree

0.874

0.0191

0.834

0.907

LR

0.825

0.0222

0.780

0.864

The performances of the investigated models for the test dataset based on the ROC curve, AUC, SE, and 95% CI values are shown in Figure 8 and summarized in Table 7. All models performed well, but the proposed new ensemble model, RS-REPTree, has the highest prediction capability based on AUC (0.860), SE (0.0315), and 95% CI (0.793-0.912). It is followed by the Bag-REPTree (AUC = 0.841), LR (AUC=0.824), AB-REPTree (AUC = 0.805), and REPTree (AUC = 0.800) models. Overall, our results show that the new ensemble models of REPTree outperform and outclass the standard REPTree model in gully erosion susceptibility mapping.

Figure 8. ROC curve and AUC of the models: (a) training dataset, and (b) validation dataset

Table 7. ROC curve using the validation dataset

Model

AUC

SE

95% CI

Lower bound

Upper bound

REPTree

0.800

0.0383

0.725

0.862

AB-REPTree

0.805

0.0368

0.731

0.866

Bag-REPTree

0.841

0.0329

0.771

0.896

RS-REPTree

0.860

0.0315

0.793

0.912

LR

0.824

0.0350

0.751

0.882

Choice of hypermeters for the ensemble algorithms.

Response:

Thank you for your attention. This comment has been repeated above. We in this study added a table that can address the hypermeters for the ensemble algorithms. We copy here for fast revision:

“Heuristic tests were used to find the best values for the four model’s parameters, which are shown in Table 3".”

Table 3 Parameters of algorithms utilized in this study.

Methods

Algorithms

Parameters

Base classifier

Reduced-error pruning tree

Seed, 1ï¼›The minimum total weight of the instances in a leaf, 2; Number of folds, 10

Ensembles

Bagging

Seed, 1; The number of iterations, 10

AdaBoost

Seed, 1; The number of iterations, 10

Random Subspace

Seed, 1; The number of iterations, 10

Stability of the proposed methods.

Response:

Thank you for the comment. We well know this concern of the reviewer. As you know, there are some machine learning ensemble models applied for landslide susceptibility models over the world that they are different results from a case study to another one because of diversity and different of geo-environmental factors. Therefore, the stability of each developed model is for the case study where the model is performed and tested there. In other words, to check the stability of the proposed model in a given study area it should be tested in another study area and the results should be compared and decision to be made.  

Round 2

Reviewer 1 Report

Thank you for the thorough consideration of my comments, and the excellent additions to the manuscript.

Reviewer 2 Report

The authors have addressed my comments and have incorporated into the revised version. I approve the manuscript in its current form.